# Breadth-First Pipeline Parallelism

**Joel Lamy-Poirier**
ServiceNow Research
Montreal, Québec, Canada
joel.lamy-poirier@servicenow.com

## Abstract

We introduce Breadth-First Pipeline Parallelism, a novel training schedule which optimizes the combination of pipeline and data parallelism. Breadth-First Pipeline Parallelism lowers the training time, cost and memory usage by combining a high GPU utilization with a small batch size per GPU, and by making use of fully sharded data parallelism. Experimentally, we observed increases of up to 53% in training speed.

## 1  Introduction

Large language models [1, 20] are quickly becoming an essential tool for natural language processing. However, a challenging aspect of developing such models is their long and expensive training process. A single training may require tens, or even hundreds of thousands of GPU-days worth of computation [1, 7, 12]. This results in price tags that can reach several million dollars and in a large environmental footprint. Significant efforts have been made towards reducing the training duration and cost, for example by improving the model [4], the training scheme [7, 9] or the hardware utilization [2, 10, 12, 13, 17]. However, the training time and cost can only be jointly optimized up to a certain point, as there is an inherent trade-off between them.

On the one hand, reducing training time requires an increased number of GPUs, which in turn needs a higher batch size. Indeed, these extra GPUs will typically be added through data parallelism, so they need to process *different* samples. In general, distributed training requires a *minimum batch size per GPU* ($\beta_{\min}$), which is equal or slightly smaller than one, depending on the method used. In practice, most models are trained with a batch size per GPU much higher than this bare minimum, to allow for a higher GPU utilization.

On the other hand, increasing the batch size hurts the effectiveness of stochastic gradient descent (SGD) [11]. While a large body of work [5, 9, 11, 15, 18] has demonstrated that large batches are able to train machine learning models, given a careful adjustment of the training hyperparameters, large batches train more slowly and require extra training samples to reach the same validation loss. That is, they add an overhead which increases the training cost (and time). This overhead is small when the batch size is well below an empirical variable known as the *critical batch size* ($B_{\mathrm{crit}}$), but increases when approaching $B_{\mathrm{crit}}$.

Thus, the trade-off can be summarized as follows: reducing the training time requires a larger batch, while reducing the training cost requires a smaller one. We stress that this concerns the entire training process rather than the batch time or GPU utilization which, while important, do not tell the full story. Although this trade-off is difficult (if not impossible) to avoid, we can mitigate it by reducing the batch size per GPU as much as possible, ideally to $\beta_{\min}$. However, there is a major obstacle to doing so: existing parallelization methods are inefficient at $\beta_{\min}$. Indeed, the state-of-the-art methods such as 2d [2, 21] and 3d [10, 12] parallelism are able to achieve a high GPU utilization (i.e., to use a high fraction of the available flop/s), but require a batch size per GPU significantly higher than $\beta_{\min}$ to do so.

Has it Trained Yet? Workshop at the Conference on Neural Information Processing Systems (NeurIPS 2022).

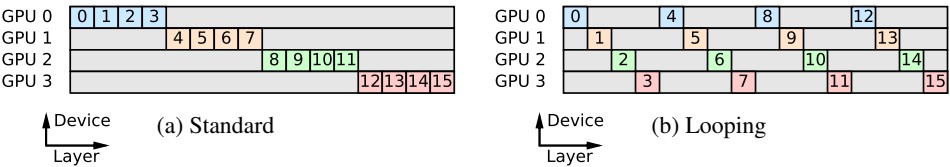

Figure 1: Standard and looping layer placements for a 16-layer model.

Therefore, to train large language models more efficiently, we should look for a training method that not only achieves a high GPU utilization, but that does so with a *low batch-size per GPU*. We propose a novel method, Breadth-First Pipeline Parallelism, that achieves precisely that. It uses a *looping* placement of the network layers, together with a *breadth-first* micro-batch schedule. Looping pipelines provide a way to reduce the pipeline-parallel overhead from the *pipeline bubble*, as opposed to the more common mitigation method of increasing the batch size. They were first introduced in [12] but remained limited in practice due the associated increase in memory and network usage. The breadth-first schedule avoids both of these limitations: it allows lowering the memory usage to a minimum with (fully) *sharded data parallelism* [13], and prevent a network bottleneck by maximizing the *overlap* between network communication and computation. Experimentally, we observed an increase in throughput of up to 53% near $\beta_{\mathrm{min}}$ for a 52 billion parameter model, which translates into a similar reduction in training cost and time reduction on large clusters.

## 2    Breadth-first pipeline

In typical pipeline-parallel setups [6, 8] each device hosts a single contiguous set of layers, or a *stage* (Figure 1a). The stages should be identical or near-identical in size, so that they take about the same time to process a micro-batch. This approach minimizes the network usage, but suffers a large overhead from the *pipeline bubble*, where some GPUs are idle as the pipeline gets filled and emptied. There are two common schedules for pipeline parallelism: with *GPipe* [8], the entire forward pass is run first (Figure 2a), followed by the backward pass, while with *1F1B* [6], the forward and backward steps are alternated so that earlier micro-batches finish as soon as possible (Figure 2b). Although the choice of schedule makes no difference in training time, the latter requires less activation memory when there are many sequential micro-batches.

By contrast, in a looping pipeline [12], we divide the model into many (identical or near-identical) stages, wrapping them around by connecting the first and last device so that each GPU is assigned multiple stages (Figure 1b). Looping the pipeline allows a given micro-batch to reach the end of the pipeline earlier, leading to a smaller pipeline bubble. With a looping pipeline, the schedule depends on one important choice: when a micro-batch is ready to wrap around the pipeline, the first device may either immediately process it or keep processing the queued micro-batches for earlier stages. The former option combines naturally with alternating forward and backward steps and may be qualified as *depth-first* (Figure 2c). It was introduced in [12] with the goal of minimizing activation memory. However, we argue that the latter, *breadth-first* option is preferable (Figure 2d).

To maximize the benefits of looping, i.e., to minimize the bubble overhead with a small batch size, there should be as many stages as possible per device. Since the total number of stages is limited by the depth of the model, this requires fewer pipeline-parallel devices. However, such *shallow pipeline* may be harmful to training due to other factors.

Pipeline parallelism is typically used to reduce the memory usage for the training state, and a shallow pipeline may run out of memory for large models. We remediate to this problem with *(fully) sharded data parallelism* [13] (see also Appendix A), in which the training state is split between the data-parallel devices. However, sharded data parallelism is generally inefficient for multiple sequential micro-batches. This is because the network operations are repeated each time a layer is involved in a computation, i.e., for each micro-batch (see Appendix C for a detailed example.). The breadth-first schedule avoids these repetitions by bundling together the computations for a given layer.

Pipeline parallelism also allows reducing the data-parallel network communication, as each device has fewer gradients to share in the reduction step.[1] Thus, while the network usage is small for deep

---

[1]We consider a non-sharded model for simplicity, but an equivalent argument holds in the sharded case.

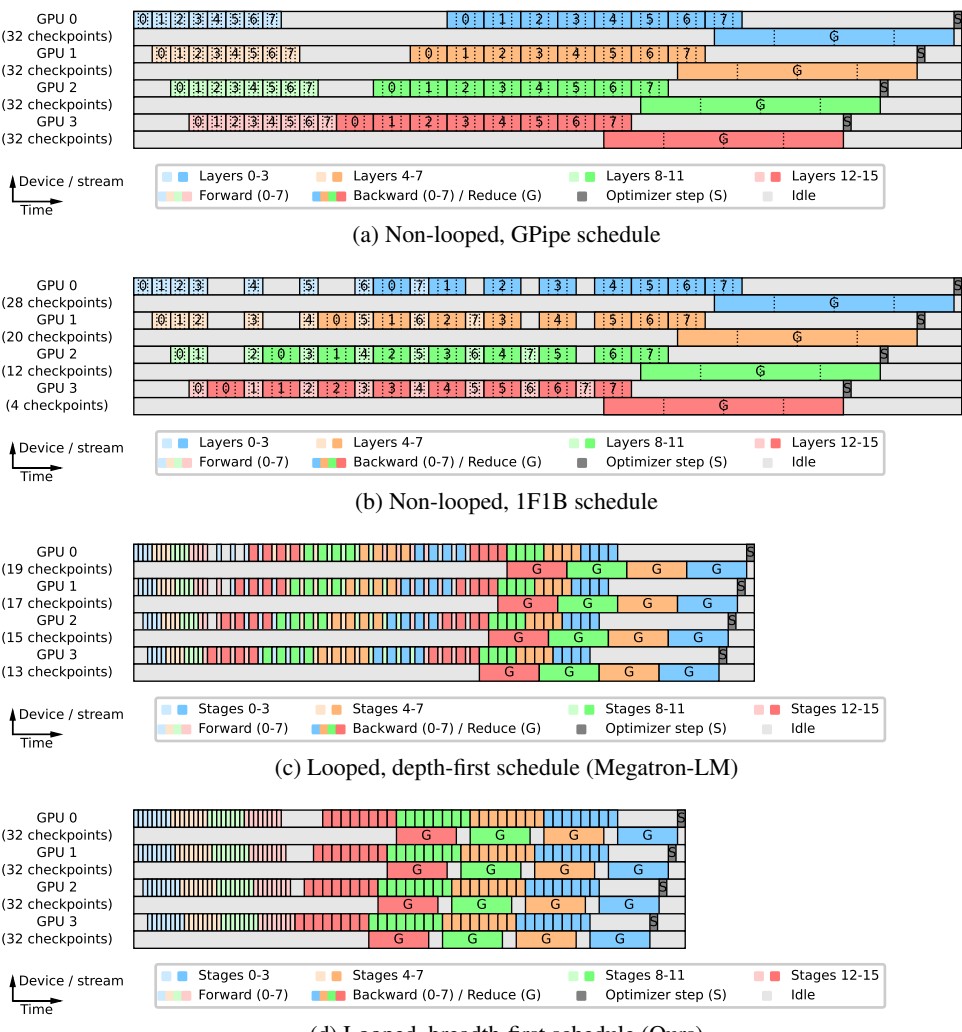

Figure 2: Comparison of the four pipeline schedules considered in this paper (times to scale), for a 16-layer model on 4 pipeline devices, with 8 sequential micro-batches (numbered 0-7), in the presence of data parallelism. We show both the computation (even rows) and the data-parallel communication (odd rows), assumed to run in parallel CUDA streams. (We omit the pipeline-parallel communication for simplicity.) Looped schedules run significantly faster than their non-looped counterparts, with breadth-first being the fastest.

pipelines [12], this is not the case for shallow ones. To preserve efficiency, it is essential that the network is fast enough, and that the operations are *overlapped* (run in parallel) with the computation. The breadth-first approach has the highest overlap of all methods, allowing to overlap with nearly all the backward pass rather than the last micro-batch(es). (The network efficiency could also be improved with a higher batch size, but this is precisely what we are trying to avoid.)

On a side note, a large number of stages leads to frequent pipeline-parallel network communication, and these will be harmful to efficiency unless overlapped with computation. Both schedules allow for such overlap but require an increase in batch size so that no device is input-starved during the transfers. For large models and fast networks, this increase can be limited to a single micro-batch.

Summing up, a *looping pipeline* removes much of the pipeline bubble and its associated overhead, with maximal benefits for *shallow pipelines* with many stages. A *breadth-first schedule* is required to limit the memory usage for larger models and to prevent an overhead from the increased network communication. The combination of the looping pipeline and breadth-first schedule, or *Breadth-First Pipeline Parallelism*, allows for training efficiently with a *low batch size per GPU*, i.e., near $\beta_{\min}$. This assumes a large model and fast network connection for efficient data and pipeline network communication

## 3 Evaluation

We ran a series of experiments to verify the following claims:

1. A breadth-first pipeline improves GPU utilization at a low batch size per GPU.

2. The large model assumption holds for a typical large language model.

3. A breadth-first pipeline reduces the time and/or cost of training large language models.

For this purpose, we evaluated the computational throughput (flop/s) for a 52 billion parameter BERT model [3] trained on a cluster of 64 V100 GPUs, for a selection of batch sizes, and four methods: breadth-first pipeline, depth-first pipeline, a non-looped pipeline, and no pipeline at all. To ensure a fair comparison, we tested a wide variety of configurations and selected the most efficient one for each batch size and method. More details on our experiment setup can be found in Appendix D.

Claim 1 is demonstrated in Figure 3a. The breadth-first approach outperforms all other methods at a low batch size per GPU, and trains 53% faster than the non-looped baseline slightly above $\beta_{\min} = \frac{1}{8}$ (with one extra sample to allow for pipeline-parallel network overlap). Only the non-pipelined method achieves a higher utilization, but for an excessively high batch size per GPU.

As for claim 2, our results show the large model assumption hold for the 52 billion parameter model, but imperfectly. The breadth-first approach achieves a fair GPU utilization of $34\%$ near $\beta_{\min}$, but does benefit from a larger batch, reaching up to $44\%$. However, this is largely due to the overhead of tensor model parallelism [16, 17], which has little to do with our method but also affects the batch size per GPU. (See detailed results in Appendix F.) This is to be expected, as for example Megatron-LM only achieves its peak efficiency above 500 billion parameters [10, 12].[2]

Verifying claim 3 required some extrapolation as we did not have access to a sufficiently large cluster, and it was not possible to run training to completion. We extrapolated our results to a range of cluster sizes, by scaling data parallelism and the batch size (proportionally), which we assumed not to affect the computational efficiency.[3] We evaluated the training time for each extrapolation for a base training length of 50,000 times the critical batch size (347 billion tokens), to which we added the overhead from the batch size (see Appendix B). We used the results of [9] to estimate $B_{\mathrm{crit}}$ ($\approx 6780$) [9]. The resulting trade-off curves are plotted in Figure 3b. The breadth-first approach shows a significantly improved trade-off curve when compared to other methods allowing for a reduced training time and cost at nearly all cluster sizes. Although the non-pipelined method allows for the lowest cost, it only does so for an impractical training time of above six months.

---

[2]We refrain from numerically comparing our results with these papers, which used a different GPU model and experiment setup.

[3]This is justified because the compute and network usages per GPU are largely unaffected by this scaling.

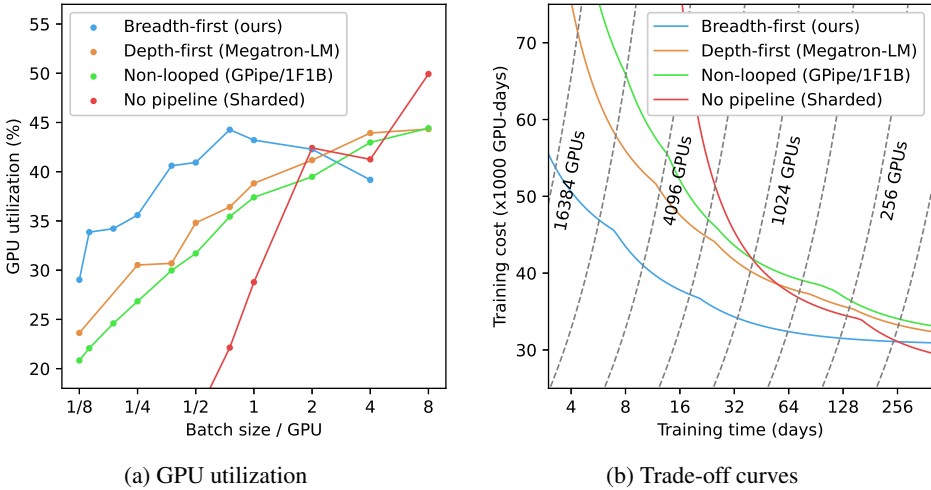

|                        |                      |
|:----------------------:|:--------------------:|
| (a) GPU utilization    | (b) Trade-off curves |

Figure 3: The GPU utilization for a 52 billion parameter model on 64 V100 GPUs as a function of the batch size per GPU for the four selected methods, and predicted trade-off curves.

## 4   Discussion and conclusion

We determined that Breadth-First Pipeline Parallelism reduces the training time and cost of training large language models by combining a high efficiency with a low batch size per GPU. As an added bonus, the method uses significantly less memory than the current alternatives (as little as 3 GB for the 52B model, see Appendix F), and should be able to train models with tens if not hundreds of trillions of parameters on current hardware, at least from a memory perspective. This was already possible with methods such as ZeRO-infinity [14], but with a prohibitively high training time. Although our method improves on that level, models of these sizes remain fundamentally limited in terms of *both* training time and cost.

Recently, a new method was proposed which avoids most activation recomputation, lowering the amount of computation by up to $25\%$ [10]. However, this comes at the cost of an increased activation memory, which is inversely proportional to the pipeline depth so represents an additional obstacle to shallow pipelines. Breadth-First Pipeline Parallelism is not only fully compatible with the method, but also improves on it since the memory usage is lower to begin with.

## Acknowledgements

The author is thankful to Harm de Vries for providing extensive support in writing the paper, and to Stefania Raimondo, Adam Salvail and Chris Tyler for reviewing and providing feedback.

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

## A   Distributed training

In this section, we review the state-of-the-art methods for training large language models.

When it comes to distributed training, data parallelism is at the core of all the state-of-the-art methods, due to its simplicity and scalability. In this setup, each device independently calculates the loss and gradients for its own micro-batch(es), then shares the results in the gradient reduction step using an all-reduce operation and updates the weights using the reduced (summed) gradients. However, data parallelism alone is limited by memory, batch size and network. It requires each device to hold a copy of the training state, which for large models is too big to fit in memory. By design, it also affects the batch size. Each device needs to process at least a single sample, but more may be required for computational efficiency. Multiple samples may be needed to fully occupy the device, i.e., to achieve good *thread-level parallelism*, unless the model is very large. Furthermore, the gradient reduction may cause a bottleneck if it takes longer than the computation, and this bottleneck is prevented by increasing the batch size (per GPU). The reduction can be *overlapped* with much of the backward pass, i.e., run in parallel with minimal overhead on either operation, so data parallelism is computationally efficient provided the reduction is faster than the overlapped computation.

The issues of data parallelism can be addressed to some extent by combining it with model parallelism – tensor and pipeline – where the model itself is divided between the devices. As each device is responsible for a fraction of the model weight, the memory and data-parallel network requirements are both significantly reduced.

In tensor model parallelism, the model is divided along its width. Each of the tensor-parallel devices processes a subset of the channels for the *same* samples. Tensor parallelism reduces the batch size per GPU, but this also means less per device, which reduces thread-level parallelism and negates the benefits of the smaller gradient reduction. There is a large amount of communication between the (tensor-parallel) GPUs, which restricts the method to the fastest intra-node networks such as NVLink, and to sufficiently large (wide) models.

In pipeline model parallelism, the model is divided along its depth. Each of the pipeline-parallel devices processes a subset of the layers for the same samples, but at *different times*. Multiple sequential micro-batches are required to achieve parallel computation, at least one per device. Thus, the minimum batch size per GPU is the same as data parallelism, and the network bottleneck may be avoided due to the reduced network communication. However, running with this batch size is still inefficient due to the *pipeline bubble*, where the GPUs remain idle half the time as the pipeline gets filled or emptied. Many more micro-batches are needed for computational efficicency, so that the GPUs are idle for a smaller fraction of the time.

Recently, a new form of data parallelism was proposed which lowers the memory usage without the need for model parallelism. In *sharded data parallelism* [13] the training state is split between the data-parallel devices, each holding and updating a small *shard* of the weights. With sufficiently many devices, the state memory usage is reduced to a minimum, although some buffers are still needed to hold local weights and gradients. In *fully sharded* data parallelism (FSDP), the local buffers only hold the weights and gradient that are currently being used, so do not require much memory. However, the layers need to be *reconstructed* before each usage with an all-gather, and the gradients are reduced as they are computed with a reduce-scatter. For a single micro-batch, there are two reconstructions (forward and backward) and one reduction for each layer, each operation taking half the time of an all-reduce, for a net increase of 50% in network usage. However, if there are multiple sequential micro-batches, the operations are repeated for each of them. In particular, the method is impractical when combined with pipeline parallelism. *Partially sharded* data parallelism (PSDP) avoids the increase in network usage by running the operations only once per batch, keeping the local weights between usages, as well as the gradients if there are multiple micro-batches. This limits the memory reduction to about 4x or 8x, which is not enough for large models but may complement other methods.

Training large language models requires a combination of distributed methods. All state-of-the-art methods rely on some form of data parallelism for its scaling properties, as well as tensor parallelism for its reduced batch size per GPU and memory usage. However, they differ in their choice between pipeline parallelism and FSDP. The most common method, *3d parallelism*, uses a combination of data, tensor and pipeline parallelism. It was for example used to train GPT-3 (175 B parameters) [1] and Megatron-Turing NLG (530 B) [19] The latter also used PSDP. An alternative method, *sharded 2d parallelism*, use a combination of FSDP and tensor parallelism. It has been successfully used to train PaLM (540 B parameters) [2] and OPT (175 B) [21]. Our method, *breadth-first* pipeline parallelism, removes this choice and allows for *both* FSDP and pipeline parallelism.

## B    Batch size

In stochastic gradient descent, a (mini-)batch is used to approximate the true gradients of the weights with respect to the loss. Increasing the batch size B generally improves this approximation, leading to more efficient steps. For small batches, this is computationally efficient, with larger batches allowing to train for proportionally fewer steps, for a near-constant computing power. However, for large batches the approximation is already accurate and additional samples provide a negligible improvement, leading to a waste of computing power.

Empirically, the number of samples needed to reach a given validation loss has been shown to follow the curve [11]

$$\text{Samples} \propto 1 + \frac{B}{B_{\text{crit}}}, \tag{1}$$

where the *critical batch size* $B_{\text{crit}}$ depends on the model, training scheme and target validation loss, and it can be estimated by measuring the gradient statistics [11]. In short, the relative overhead is equal to the ratio $B/B_{\text{crit}}$. For example, GPT-3 was trained with a batch size of 3 million tokens, with a critical batch size estimated to 10 million tokens [9], for an overhead of about 30%.

For both state-of-the-art methods, the number of GPUs used for training ($N_{\text{GPU}}$) can be scaled with minimal impact on the GPU utilization, mainly with data parallelism, provided the batch size per GPU is kept constant. Along the associated trade-off curve, the batch size and number of training steps are good proxy variables for the number of GPUs and the training time, respectively. The cost, on the other hand, is roughly proportional to the total number of samples processed. Therefore, the trade-off curve can be approximated by

$$\text{Cost} \propto 1 + \beta \frac{N_{\text{GPU}}}{B_{\text{crit}}}, \qquad \text{Time} \propto \frac{\text{Cost}}{N_{\text{GPU}}}, \tag{2}$$

## C    Breadth-first gradient accumulation

We consider a data-parallel scenario with multiple sequential micro-batches. This may happen when a high batch size is needed to mitigate the gradient reduction network overhead, and the micro-batch size is limited by activation memory constraints. In that case, we typically use a *depth-first* schedule, where a given micro-batch goes through the entire forward and backward passes before the next one begins. This schedule achieves the goal of limiting the memory usage, as all intermediate activations can be dropped between micro-batches. However, the gradient reduction cannot begin until the last micro-batch, leading to poor overlap with computation (Figure 4c). Therefore, the network overhead is mitigated, but not eliminated. In the (fully) sharded data-parallel case, there is no mitigation at all since the network operations (reconstruction and reduction) need to be repeated for each micro-batch (Figure 4b).

A breadth-first schedule solves these problem, allowing to overlap the gradient reduction with most of the backward pass, and in the sharded case avoiding duplicating the operations (Figures 4c and 4d). The breadth-first schedule comes at the cost of memory, since more activations need to be stored at once. However, when using activation checkpoints, this memory increase remains small, and for larger models the memory savings from sharded data parallelism are far more important.

However, when the stage outputs coincide with activation checkpoints, this only increases the checkpoint memory, which remains smaller than the intra-stage activation unless there are lots of sequential micro-batches. Furthermore, for large models, the state memory is the bottleneck, so

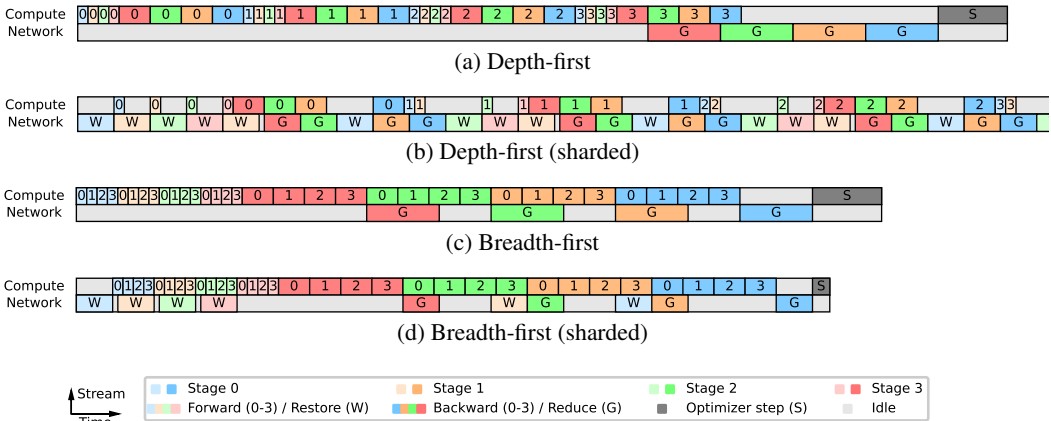

Figure 4: Example depth-first and breadth-first schedules for gradient accumulation with standard and sharded data parallelism. The depth-first approach achieves poor network overlap, and in the sharded cases involves a costly repetition of the network operations. Both issues are solved with the breadth-first schedule, resulting in a faster training.

the memory usage may be *lower* for the breadth-first schedule when combined with sharded data parallelism.

# D    Implementation

We implemented Breadth-First Pipeline Parallelism on our own, custom library. As a reference, we used the source code for Megatron-LM as it was just before the publication of [10] (commit e156d2f). We verified that the model forward and backward passes are identical, generating the same kernels on the GPU (other than the modifications listed here). Although we tried our best to compare methods rather than implementations, our codebase does not support the depth-first/1F1B schedules, for which we used Megatron-LM instead. This may skew the results for these configurations, as Megatron-LM does not support FSDP or PSDP,[4] does not overlap network communication, and only support certain batch sizes with the depth-first (interleaved) schedule.

## D.1    Distributed training

We used a custom implementation of data parallelism, with builtin support for mixed precision and sharded data parallelism. It relies on the breakdown into stages to optimize the data conversion and transfer, where the stage size may be set to any number of transformer layers. For that purpose, the embedding and final layers are either treated equivalently as transformer layers, or merged with the adjacent layer, depending on what is most efficient for a given configuration. We use a double-buffered approach to achieve network overlap at a minimal memory cost. For example, the computation for a given stage may be done in parallel with the weights for the next stage being restored on the other buffer. The network operations are performed in-place, avoiding the memory and kernel time overhead of network buffers.

We implemented Breadth-First Pipeline Parallelism, with support for network overlap as described in section 2. It reduces to standard, non-looped pipeline parallelism when the stages are sufficiently large.

## D.2    Memory efficiency

Large models tend to suffer heavily from memory fragmentation, where the GPU has enough memory to allocate a given tensor but not in a contiguous chunk, which leads to unnecessary out-of-memory errors. To reduce the fragmentation, we pre-allocate tensor whenever possible, including for the

---

[4]Megatron-LM added support for PSDP ("distributed optimizer") in a later version, published alongside [10], but it could not be included in our experiments which were already underway.

training state (fp32 weights and gradients, optimizer momenta), fp16 weight and gradient buffers, the activation checkpoints, the pipeline receive buffers. Apart from a few small tensors, this leaves the intermediate layer activations and their gradients as the only actively allocated tensors; these still suffer heavily from memory fragmentation but are difficult to pre-allocate within Pytorch.

We also observed a significant memory overhead and in some cases important slowdowns due to how the Pytorch caching allocator is implemented.[5] The tensor-parallel network operations are run in a separate CUDA stream set up by NCCL, and while that stream is immediately synchronized with the default stream, this limits the ability to free the tensors involved in the operation. As a result, the tensor memory is blocked from the CPU perspective until the operation is completed on GPU, which increases the memory usage when there are many queued kernels.[6] This may prevent the caching allocator from finding enough memory for future kernels, at which point it synchronizes the GPU, then flushes the cached allocations.[7] The flushing operation is relatively slow, causing some idle time on the GPU side. The overhead is multiplied when there are many parallel devices, as the slowdowns happen at different times, and each is enough to block the whole group. In some cases, we observed a combined overhead of more than 100%. We fixed this by explicitly preventing the kernel queue from growing two big, by adding frequent CUDA synchronizations (on earlier events, so the synchronization is non-blocking.)

## E    Experimentation details

We tested with a 52 billion parameter Bert model, with an architecture and training scheme identical to the Megatron-LM implementation [12, 17]. The model consists of 64 transformer layers consisting of 62 attention heads of size 126, for a hidden size of 8192. We used a sequence length to 1024 for all our experiments. We measured the computational throughput by evaluating the batch time, averaged over 50 batches. We evaluated the computational throughput from the the sequence length ($S$), the hidden size ($H$), the number of layers ($N$) and the vocabulary size ($V$), using the formula [12]

$$\text{Compute per token} \approx 96NH^2 \left(1 + \frac{S}{6H} + \frac{V}{16NH}\right).$$

We approximated the critical batch size $B_{\text{crit}} \approx 6780$ using the results of [9], which suggest the approximation

$$B_{\text{crit}} \approx 2.02e3 \times P^{0.33}/S,$$

where $P$ is the number of parameters in the model.

We ran our experiments on a cluster of eight DGX-1 servers, for a total of 64 V100-SXM2-32GB GPUs, connected through an Infiniband network. All our experiments were run on the same hardware (and software), except for one node which was temporarily replaced due to a hardware failure.

## F    Detailed results

The most efficient configuration for each method and batch size is listed in table 1.

---

[5]This problem is not unique to our implementation. It was observed with *Pytorch Fully Sharded Data Parallel*, and we were able to reproduce it on Megatron-LM without pipeline parallelism. (The Megatron-LM implementation of pipeline parallelism includes frequent CUDA synchronizations which, prevent the issue from happening, although inefficiently.)

[6]In that scenario, some of the tensors involved may have been deleted on CPU, which means the underlying memory block will be available when the queued kernels complete. The CUDA caching allocator provides a way to reuse that memory before the kernels are run, by predicting the future memory usage. However, it can only do so efficiently in a single-stream setting.

[7]The flushing is designed for a different scenario, where the memory is available but there is no cached block of the correct size. In the present case, the flush is generally unnecessary as the synchronization frees up many blocks but is performed either way.

Table 1: Selected optimal configurations for the 52 billion parameter model.

| Method | Batch size | Implementation | Pipeline parallel | Tensor parallel | Micro-batch size | Sequential micro-batches | Stages per device* | Sharded | Throughput (Tflop/s/GPU) | Memory** (GB) |
|---|---|---|---|---|---|---|---|---|---|---|
| Breadth-first | 8 | Ours | 8 | 8 | 1 | 8 | 4 | ✗ | 36.28 | 15.96 |
| Breadth-first | 9 | Ours | 8 | 8 | 1 | 9 | 8 | ✗ | 42.33 | 14.74 |
| Breadth-first | 12 | Ours | 8 | 8 | 1 | 12 | 4 | ✗ | 42.77 | 16.66 |
| Breadth-first | 16 | Ours | 4 | 8 | 1 | 8 | 8 | ✓ | 44.49 | 16.60 |
| Breadth-first | 24 | Ours | 4 | 8 | 2 | 6 | 8 | ✓ | 50.76 | 17.96 |
| Breadth-first | 32 | Ours | 8 | 4 | 1 | 16 | 4 | ✓ | 51.17 | 19.12 |
| Breadth-first | 48 | Ours | 8 | 2 | 1 | 12 | 8 | ✓ | 55.34 | 19.73 |
| Breadth-first | 64 | Ours | 4 | 2 | 1 | 8 | 16 | ✓ | 54.01 | 20.23 |
| Breadth-first | 128 | Ours | 4 | 2 | 2 | 8 | 8 | ✓ | 52.85 | 24.65 |
| Breadth-first | 256 | Ours | 2 | 2 | 1 | 16 | 32 | ✓ | 48.97 | 26.32 |
| Depth-first | 8 | Megatron-LM | 8 | 8 | 1 | 8 | 2 | ✗ | 29.53 | 15.78 |
| Depth-first | 16 | Megatron-LM | 8 | 8 | 2 | 8 | 4 | ✗ | 38.16 | 15.94 |
| Depth-first | 24 | Megatron-LM | 8 | 8 | 1 | 24 | 2 | ✗ | 38.37 | 15.78 |
| Depth-first | 32 | Megatron-LM | 8 | 8 | 4 | 8 | 4 | ✗ | 43.50 | 17.77 |
| Depth-first | 48 | Megatron-LM | 8 | 8 | 2 | 24 | 2 | ✗ | 45.52 | 16.27 |
| Depth-first | 64 | Megatron-LM | 8 | 8 | 4 | 16 | 4 | ✗ | 48.52 | 19.18 |
| Depth-first | 128 | Megatron-LM | 8 | 8 | 4 | 32 | 4 | ✗ | 51.46 | 19.18 |
| Depth-first | 256 | Megatron-LM | 16 | 4 | 4 | 64 | 2 | ✗ | 54.91 | 21.35 |
| Depth-first | 512 | Megatron-LM | 8 | 8 | 4 | 128 | 2 | ✗ | 55.41 | 19.87 |
| Non-looped | 8 | Ours | 8 | 8 | 1 | 8 | 1 | ✗ | 26.04 | 16.87 |
| Non-looped | 9 | Ours | 8 | 8 | 1 | 9 | 1 | ✗ | 27.59 | 16.99 |
| Non-looped | 12 | Ours | 8 | 8 | 1 | 12 | 1 | ✗ | 30.74 | 17.38 |
| Non-looped | 16 | Ours | 8 | 8 | 1 | 16 | 1 | ✗ | 33.54 | 17.89 |
| Non-looped | 24 | Ours | 8 | 8 | 1 | 24 | 1 | ✗ | 37.46 | 18.91 |
| Non-looped | 32 | Ours | 8 | 8 | 2 | 16 | 1 | ✗ | 39.62 | 20.12 |
| Non-looped | 48 | Ours | 8 | 4 | 1 | 24 | 1 | ✓ | 44.30 | 22.71 |
| Non-looped | 64 | Ours | 8 | 4 | 1 | 32 | 1 | ✓ | 46.74 | 23.75 |
| Non-looped | 128 | Megatron-LM | 8 | 8 | 2 | 64 | 1 | ✗ | 49.35 | 15.75 |
| Non-looped | 256 | Megatron-LM | 16 | 4 | 2 | 128 | 1 | ✗ | 53.72 | 16.33 |
| Non-looped | 512 | Megatron-LM | 8 | 8 | 4 | 128 | 1 | ✗ | 55.52 | 17.68 |
| No pipeline | 8 | Ours | 1 | 8 | 1 | 1 | 1 | ✓ | 4.73 | 14.23 |
| No pipeline | 16 | Ours | 1 | 8 | 2 | 1 | 1 | ✓ | 9.43 | 15.44 |
| No pipeline | 24 | Ours | 1 | 8 | 3 | 1 | 1 | ✓ | 14.07 | 16.64 |
| No pipeline | 32 | Ours | 1 | 8 | 4 | 1 | 1 | ✓ | 18.79 | 17.85 |
| No pipeline | 48 | Ours | 1 | 8 | 6 | 1 | 1 | ✓ | 27.66 | 20.29 |
| No pipeline | 64 | Ours | 1 | 8 | 8 | 1 | 1 | ✓ | 35.97 | 22.73 |
| No pipeline | 128 | Ours | 1 | 2 | 4 | 1 | 1 | ✓ | 53.01 | 21.43 |
| No pipeline | 256 | Ours | 1 | 2 | 4 | 2 | 1 | ✓ | 51.57 | 21.43 |
| No pipeline | 512 | Ours | 1 | 2 | 4 | 4 | 1 | ✓ | 62.40 | 21.44 |

* In the breadth-first case, this excludes the embedding and output layers, which are treated as separate layers and add an extra stage to the first (and second in some cases) device.

** For sharded configurations, this is not representative of the memory usage on larger clusters, which would about 12 GB lower.

