# OpenReview forum: "Breadth-first pipeline parallelism"
_NeurIPS.cc/2022/Workshop/HITY — HITY Workshop NeurIPS 2022_

### Official Review · Reviewer_pN2q · 2022-10-08
**Efficient distributed training scheme for small batch sizes**

**Rating:** 1
**Confidence:** 2

**Review:**

Training large DNNs with stochastic optimizers is effective at relatively small batch sizes. Existing techniques to distribute training across many devices achieve high device utilization, but also lead to an increased effective batch size that yields diminishing returns of the optimization. This paper studies a new schedule which features high device utilization at small batch sizes. This is demonstrated on a 52 billion parameter model on 64 V100 GPUs by comparison to other schemes.

---

Miscellaneous comments:

- L89: Missing space between "models" and "and"
- L252: "we use" → "uses"

---

### Official Review · Reviewer_4nMA · 2022-10-19

**Rating:** 1
**Confidence:** 3

**Review:**

Interesting parallelism setup for small batch sizes. Some nits:

- "In general, distributed training requires a minimum batch size per GPU (βmin), which is generally close to one." why is this the case? many pipelines train with >>1 example per GPU.
- "On the other hand, increasing the batch size hurts the effectiveness of stochastic gradient descent" the citation you include for that does not say that larger batches always hurt the effectiveness. in fact, many papers have shown that large batches train just fine with standard optimizers https://arxiv.org/abs/2102.06356 https://arxiv.org/abs/1811.03600

---

### Official Review · Reviewer_JPa3 · 2022-10-20

**Rating:** 0
**Confidence:** 4

**Review:**


**Summary**

This submission focuses on the efficiency of gradient computations a distributed system. Compared to alternative schemes, the proposed method increases the throughput in distributed settings with low per-gpu batch size by increasing memory requirements or communication costs through the use of sharding.

**Workshop fit**

This is crucial in the training of large-scale state-of-the-art systems. The topic is of interest to practitioners and would be a good fit for the NeurIPS general audience. For a HPC/System-for-ML workshop (not running this year), my recommendation would be a clear accept. However, it does not seem to be a good fit for this workshop. as its stated focus is explicitly on algorithmic rather than hardware or software developments.

**My understanding of the goal of the workshop may be incomplete and I encourage the organizers to accept this paper if they wish to include system-focused research.**

---

**Detailed comments**

For the authors, my main feedback is that the current writing is aimed at experts in the specific community of distributed systems for deep learning. It would benefit from an introduction aimed at a general audience. A reminder of community specific terminology, such as `micro batch`, would help. An example of the order of magnitude of the system would help grasp the type of problems the paper is dealing with (eg “consider training model X with 16 machines each with 4 GPUS”), as more methodology or theory-focused communities typically only use single-GPU machines.

Some examples of things that confused me to the point of stopping my reading;

- “distributed training requires a minimum batch size per GPU (βmin), which is generally close to one.” but it is later stated that “beta_min = 1/8” (L105). I do not understand how the system could feed only 1/8 of a sample to a GPU
- It wasn’t clear to me until reading McCandlish et al. ([10] in the submission) what the “reduced effectiveness of SGD” referred. I choose to understand it as reduced per-sample progress in the training error due to diminishing returns in variance reduction. But the focus on validation loss in the paragraph led me initially to believe it was about the reduction of generalization performance due to reduced variance. Although similar, those are different behaviors and should be separated.
- The description of the proposed could be improved. For example, In Figure 2, I only figured out that the numbers 0-3 are micro-batches identifiers by looking at the equivalent of Figure 2 in the Megatron-LM/GPipe paper (Figure 3, [arxiv.org/pdf/2104.04473.pdf#page=3](http://arxiv.org/pdf/2104.04473.pdf#page=3)).
- I do not understand the equation in Appendix B. It concerns the number of samples required to reach the critical batch size and reads `samples \propto 1 + batch size / critical batch size`. The difference is between the number of `samples` and the `batch size` is unclear. The use of proportionality confusing, as it is equivalent to `samples \propto critical batch size + batch size`. I did not find the equivalent equation in [10].

---

### Decision · Program_Chairs · 2022-10-20

Accept